# California WIC Participants Report Favorable Impacts of the COVID-Related Increase to the WIC Cash Value Benefit

**DOI:** 10.3390/ijerph191710604

**Published:** 2022-08-25

**Authors:** Catherine E. Martinez, Lorrene D. Ritchie, Danielle L. Lee, Marisa M. Tsai, Christopher E. Anderson, Shannon E. Whaley

**Affiliations:** 1Division of Research and Evaluation, Public Health Foundation Enterprises (PHFE) WIC, a Program of Heluna Health, Irwindale, CA 91706, USA; 2Nutrition Policy Institute, Division of Agriculture and Natural Resources, University of California, Oakland, CA 94607, USA

**Keywords:** WIC, diet, fruit and vegetables, diet quality, CVB, satisfaction, WIC foods, WIC benefits, nutrition, food cost

## Abstract

The United States Department of Agriculture approved an increase to the Cash Value Benefit (CVB) for the purchase of fruits and vegetables issued to participants receiving an eligible Special Supplemental Nutrition Program for Women, Infants, and Children (WIC) food package. In order to understand satisfaction, perceptions, and the overall impact of additional benefits for fruits and vegetables at the household level, a qualitative study consisting of structured phone interviews was conducted with families served by WIC in Southern California from November to December 2021 (n = 30). Families were selected from a large longitudinal study sample (N = 2784); the sample was restricted by benefit redemption and stratified by language and race. WIC participants were highly satisfied with the CVB increase, reporting increased purchasing and consumption of a variety of fruits and vegetables. Respondents noted the improved quality and variety of fruits and vegetables purchased due to the increased amount. Findings are expected to inform policy makers to adjust the CVB offered in the WIC food package with the potential to improve participant satisfaction and increase participation and retention of eligible families with benefits from healthy diets supported by WIC.

## 1. Introduction

The economic impacts of the COVID-19 pandemic are continuing to affect families throughout the United States two years after the pandemic’s inception. Access to food was interrupted, food prices increased, and food insecurity for households with children increased from 13.6% in 2019 to 14.8% in 2020 [1]. In response to the pandemic, the United States Department of Agriculture (USDA) allowed several temporary changes to the Special Supplemental Nutrition Program for Women, Infants and Children (WIC). In 2021, the American Rescue Plan Act (ARPA) passed by the United States Congress gave USDA the authority to make temporary changes to assistance programs, such as WIC, during the federally declared COVID-19 public health emergency [2]. WIC provides nutritious foods, nutrition education, breastfeeding support and social service referrals to qualifying women, infants and children up to age 5 years who reside in low-income households. Before the implementation of ARPA changes, WIC families received USD 9 for children ages 1–4 and USD 11 for pregnant and postpartum women to purchase fruits and vegetables with the cash value benefit (CVB) portion of their package [2]. One COVID-related change to WIC brought about by ARPA involved a 4-month increase to the monthly WIC CVB for fruits and vegetables (FV) to USD 35 for all WIC participants during the summer of 2021 [2]. The CVB amount was later modified for one year beginning on 1 October 2021, to USD 24 per month for children (and USD 43 or USD 47 per month for women). These revised amounts are consistent with the recommendations from the National Academies of Science, Engineering and Medicine to support half of the recommended intake of FV for children in the Dietary Guidelines for Americans [3,4,5]. Because WIC benefits are issued to WIC participants in California electronically and the augmented CVB amount was automatically loaded onto the participants’ electronic WIC card, they could begin purchasing FV as soon as the changes were in effect.

While the increased amount for the CVB has been provided, there has also been an increase in prices for all foods due to the pandemic and inflation [1]. According to the Bureau of Labor Statistics, food (at home) prices in the Los Angeles area have increased 9.5% from March 2021 to March 2022, partly driven by an 11.7% price increase for fruits and vegetables [6]. The WIC food packages are considered largely cost insensitive, since most WIC food benefits are redeemable for a specific amount of a food or beverage regardless of food prices. From 1990 to 2016, the average cost of the WIC food package per participant decreased by 23% after adjusting for inflation [7,8]. The USDA permits a 3% buffer for food package expenditures for states to account for food inflation above historical trends or unexpected occurrences such as a pandemic [8]. This allows families to continue buying the same amount of most foods with their WIC package (e.g., 16 oz loaf of bread, 1 gallon of milk) regardless of the retail price [9,10]. However, this is not the case for the CVB portion of the WIC food package, which is the only component of the package that is sensitive to fluctuations in prices because a set dollar amount is allotted per participant, as opposed to a fixed-unit benefit [11].

The CVB is an important component of the WIC food package. Compared to other food package benefits, it provides the most choice because it allows families to choose from a wide range of FV tailored to their own family or cultural preferences. FV is the food group most under consumed in the U.S. relative to recommendations, despite their many health benefits [5]. Because the USDA is considering making the CVB increase a permanent component of the WIC food packages, it is important to understand WIC participants’ views on this change [3]. The purpose of this study is to understand how the satisfaction and use of the CVB changed for WIC participants.

## 2. Materials and Methods

This qualitative study is part of a larger ongoing study to evaluate the expansion of the monthly WIC CVB in children ages 1–4 years. In the larger study, baseline survey data were collected from caregivers with children served by WIC in Southern California in April-May 2021 (N = 2784), prior to the national increase to the value of the WIC CVB, August-September 2021 (N = 1673), following the implementation of the USD 35 monthly CVB for all children, and April-May 2022 (N = 1063), following the implementation of the USD 24 monthly CVB for all children. The larger study included an oversampling of African American (AA) participants as data have shown that WIC food benefit redemption among this population is lower than other racial ethnic groups [12,13]. The focus on the AA population in this qualitative study could help to better understand potential barriers to use of the CVB and other factors affecting satisfaction with the WIC program among AA participants. The qualitative sample (N = 30) for the current study was drawn from the longitudinal sample. All aspects of the study were reviewed and approved by the California Department of Health and Human Services and the University of California, Davis Institutional Review Boards. Participants in the qualitative sample were each mailed a USD 50 gift card after completing the interview.

### 2.1. Sampling Method

Southern California WIC participants are approximately 80% Hispanic and previous research in California WIC populations has shown differences in language preference which led to the sample stratification by language [14,15,16]. This sample was also stratified by race, as the longitudinal study oversampled the AA population to better understand the differences in CVB use among this underrepresented population. Among the sample of caregivers who had completed surveys both before and after the CVB change (N = 1673), respondents were stratified by the primary language of survey (English or Spanish); the English-Speaking sample was further stratified by race (AA and non-AA). The sample was then restricted to high (top 25th percentile) and low (bottom 25th percentile) CVB redeemers and stratified by percent of the augmented CVB redeemed (high: ≥96%; low: <56%). To ensure diverse perspectives, a random sample of participants were selected from the 6 strata: Spanish-speaking, high redemption; Spanish-speaking, low-redemption; English-speaking AA, high redemption; English-speaking AA, low-redemption; English-speaking non-AA, high-redemption; English-speaking non-AA, low redemption. Recruitment by text message continued until 5 interviews were scheduled and completed for each strata. Interviews were scheduled and completed from November to December 2021. Forty participants were recruited to achieve a sample of 30 interviews (75% response rate).

### 2.2. WIC Participant Surveys and Interviews

Information about the 30 qualitative study participants was collected from three sources. Demographic information about each participant (name, date of birth for each child, household food security) was captured from the longitudinal surveys. Additional demographic information was captured from WIC administrative data (race and ethnicity, maternal education level, household size) and the qualitative interviews.

WIC staff recruited the 30 caregivers for the study and informed study participants that they would be contacted by University of California researchers to complete the interviews. Each structured interview took between 16 and 30 min and was conducted by phone by one of two researchers at the UC Nutrition Policy Institute. A female researcher with a Masters in Public Health and Registered Dietitian Nutritionist credential conducted the English interviews (D.L.L.). A male researcher with a Masters in Science conducted the interviews in Spanish. Only the interviewer and participant were on the call during the interview. Researchers informed participants that they wanted to learn about their experiences with WIC foods with the goal of using their feedback to improve the WIC program. Questions included shopping for WIC-eligible FV, preferences on amount of CVB, satisfaction with the increases to USD 35 and USD 24 per month, how much of the increase was used and whether it was hard to spend the full amount, and perceptions of whether the increase resulted in impacts on child intake of FV (both amount and variety), diet of others in the household, and ability to purchase other foods or household necessities (see Abbreviations for interview guide). The interview questions were programed into Qualtrics (Qualtrics, 2020) and interviews were recorded for transcription and for Spanish interviews, translation, and field notes were documented after the interview. Transcripts were not returned to participants for their review or comment.

Using an immersion/crystallization approach, after reading all interview transcripts, transcriptions were coded into themes using Microsoft Excel version 16.61.1, first by question and then across questions [17]. One researcher (LDR) completed the initial coding to develop the following thematic categories: (A) Experiences with CVB at USD 9 per month, (B) Experiences after CVB increase to USD 35 per month, (C) Other changes on spending outside of WIC benefits with CVB increase to USD 35 per month, (D) Impacts of USD 35 per month CVB increase on dietary intakes and (E) Experiences after CVB change to USD 24 per month. Another researcher (DLL) reviewed the transcripts and coding. Any discrepancies were discussed prior to finalizing the coding and themes. Quotes were then selected to illustrate each theme. SP after a quote indicates it came from an interview conducted in Spanish and EN signifies an interview conducted in English; the subsequent number refers to a unique caregiver. Participants did not provide feedback on the coding.

## 3. Results

### 3.1. Participant Characteristics

All respondents were women caregivers with children 1–4 years old on WIC. A majority were Hispanic (n = 20), had experienced household food insecurity in the prior month (n = 17), and had more than one child living in their household (n = 22). Families had been on WIC for an average of nearly 6 years and the average age of their youngest child on WIC was 2.7 years (range 1.3 to 3.7 years) (Table 1).

### 3.2. Major Themes

The thematic analysis resulted in five major thematic categories capturing the individuals’ experiences and perceptions that the increased CVB had on their satisfaction, purchasing changes, and diet. Perceptions and satisfaction with CVB were analyzed for each of the USD 9, USD 35 and USD 24 CVB amounts during the participant interview. Within each category, there are overarching themes for English and Spanish participants, which are summarized in Table 2 Thematic analysis was used to group the qualitative data into the five major thematic categories, and emblematic quotations are presented for each theme in Table 2.

### 3.3. Theme Category A: Experiences with CVB at USD 9 per Month

#### 3.3.1. Shopping for FV Using WIC Benefits Is Easy

The interviewed caregivers shopped for groceries an average of 1.6 times per week. They typically shopped for WIC foods at large stores and/or WIC-only stores, which are smaller stores that cater to WIC clients by providing only WIC-eligible options. All found it very or somewhat easy to use their WIC benefits to purchase the FV that they wanted for their WIC-participating child (Table 1). Those who shopped at large grocery stores noted that there was a large selection of FV to choose from in the produce section. Several cited using WIC shelf labels, the WIC mobile application, or knowledge from being on WIC in the past to help them know which FV were WIC-eligible. Shoppers at WIC-only stores described how FV access was straightforward because all items in the store are WIC eligible.

#### 3.3.2. CVB at USD 9 per Month Is Inadequate

The majority (n = 28) of caregivers interviewed reported that USD 9/month for FV was not enough; only 2 thought this amount was just right, and none thought it was too much. A common sentiment was that USD 9/month was not enough to buy all the FV that their children aged 1–4 years wanted to eat or should eat to be healthy. Many mentioned that USD 9 was enough only for about one week’s worth of FV each month. Many noted that because of the high cost of FV, they had to use their own limited funds to purchase enough FV for their child when the CVB was USD 9 per month.

### 3.4. Theme Category B: Experiences after CVB Increase to USD 35 per Month

#### 3.4.1. Satisfaction Increased with CVB Increase to USD 35 per Month

When asked about the CVB increase to USD 35 per month, the majority (n = 25) of caregivers felt it was the right amount, while three said it was not enough, and two thought it was too much for their child. Compared to USD 9, USD 35 per month allowed them to purchase FV for their child for the entire month. Many reported that the CVB increase improved the overall diet quality of their child, as caregivers could afford to be less reliant on inexpensive, more processed foods.

“I thought (the CVB increase) was amazing, because we eat more FV since there’s been an increase. (Before the increase) we were eating more processed food and I don’tthink that my daughter was getting adequate nutrition from eating those types of foods and she’s a picky eater too.”(English speaking participant with 2.5 years of WIC participation—EN4)

The few who thought that USD 35/month was not enough cited the high cost of produce while those who thought USD 35/month was too much worried about food waste.

“By the fourth week of the month, I still had a balance, meaning that I had already gotten what I needed. So I felt like if I was to use it, the fruit would just go to waste.”(English speaking participant with 7 years of WIC participation—EN6)

When asked how easy or hard it was to spend the full USD 35/month, a majority rated it as very (n = 22) or somewhat (n = 6) easy; only two said it was somewhat hard. One of the two who said it was hard found it difficult to spend the entire USD 35 in a month because her child did not eat that much. The other had difficulty getting to the store often enough to spend the USD 35 completely. The majority, however, were grateful to be able to afford more FV. Many cited that having a higher CVB amount allowed caregivers to better plan their shopping and menus.

“It (CVB increase) was easier to choose from the variety that the supermarket had without thinking about what I’m going to spend for a small amount.”(Spanish speaking participant with 13 years of WIC participation—SP31)

“I just didn’t feel pressured to go shopping and if I didn’t spend everything, the next time I went I could buy it and the fruit and everything would not spoil.”(Spanish speaking participant with 4 years of WIC participation—SP28)

“It (CVB increase) gave me more options to buy more and less worry to have... Okay if I needed to get it out of pocket am I going to have enough. So I don’t have to worry about it for the month, so if I didn’t want to spend the entire (USD) 35 in one week or you know in one day it’s like okay... Unlike with the (USD) 9,…I know it’s all going to go today.”(English speaking participant with 8 years of WIC participation—EN19)

When asked how much of the USD 35 each month participants spent on FV, respondents reported using either all (n = 24) or most (n = 6); none reported using only some or none of the increased CVB amount. Those who spent most, but not all, of the USD 35 each month spoke about challenges with getting to the grocery store at the end of the month or with getting FV purchases to add up to USD 35 exactly, resulting in a small balance remaining.

#### 3.4.2. CVB Increase Resulted in Purchasing Greater Amounts and Variety of FV

A majority said that the increase in CVB from USD 9 to USD 35/month changed their fruit (n = 27) and vegetable (n = 25) purchases. The majority said that they bought both a greater amount (n = 27) and a greater variety (n = 25) of FV. The ability to buy a greater variety of FV also enabled caregivers to introduce their child to new FV. Introducing children to new FV was more of a barrier when they received only USD 9/month because they did not want to spend such limited funds on produce that their child might not like. A positive impact reported by caregivers was that children learned to like new FV that they had been unable to try before.

“I started looking up different recipes for different vegetables because I would just buy the basics like carrots, celery, the same old stuff and make the same old soups (before the increase)… and now I buy more cauliflower, squash. We eat a lot of roasted vegetables now and I just want to expose them to a bigger variety.”(English speaking participant with 2.5 years of WIC participation—EN4)

“I thought the change was extremely helpful. Because instead of just grabbing a few breakfast fruits, I was able…to try different fresh vegetables that normally I wouldn’t try. It (CVB increase) made me go more outside my comfort zone. It actually paid off because my son ate more of the newer things than the same old stuff he was used to.”(English speaking participant with 2.5 years of WIC participation—EN13)

In addition to greater amounts and variety of FV, participants reported being able to afford a better quality FV—making eating FV more enjoyable. Participants were especially grateful during the summer when the variety of FV available was more plentiful.

“Because it (CVB increase) was like a kid in the candy store. It’s like, wow, we have this resource now that we can get more and more. They (the children) just love...They were introduced to artichokes, and they wanted dragon fruit and things like that we were not able to do before. So it was it was great.”(English speaking participant with 3 years of WIC participation—EN10)

Finally, the CVB increase to USD 35/month helped offset the cost of the increase in the amount and variety and improved quality of FV purchased for children to eat. It allowed participants the ease of not worrying about the cost of the FV, which unlike other WIC foods, are price-sensitive for WIC participants.

“We didn’t worry about the sale prices at that point… having a little bit more to work with made it where you’re not budgeting as hard for the things that you like.”(English speaking participant with 2 years of WIC participation—EN11)

## 4. Discussion

Changing the CVB component of the WIC food package provided WIC families with additional resources to reduce the burden of how much they spent for FV and minimize any detrimental impacts on diet quality associated with the COVID-19 pandemic. Prior to the CVB changes, one study in Massachusetts conducted a qualitative analysis on caregiver perspectives of the WIC program and factors that led to underutilization, finding that for many the FV benefit was the main reason for continued enrollment but, the set amount of USD 9/USD 11 per month was insufficient [18]. This study is among the first to evaluate the impact that two changes to the CVB amount (USD 35 in June 2021 and adjustment to USD 24 in October 2021) have had on satisfaction, purchasing habits, and diet quality for WIC families. A qualitative approach for this study allowed the 30 WIC participants interviewed to openly share their perceptions and experiences with the changes to the CVB amount. The restriction by percent of CVB redeemed and stratification by race/ethnicity and language yielded a broad perspective of experiences.

Based on the findings across all groups, satisfaction with the WIC food package increased with both CVB amount changes (USD 35 per month for 4 months and USD 24 per month thereafter). Overall, caregivers felt that they were able to purchase greater quantities and varieties of FV for their children that were more aligned with recommendations for health. Caregivers also noted that this allowed them more opportunities to introduce their children to a greater variety and quality of FV that were out of reach when the CVB monthly amount was set at USD 9. With the increased amount of CVB, the findings also showed that families had the ability to provide healthier options to their children during meals and snack times. With the increased amount, families had more FV access throughout their monthly benefit period instead of only the first week of their benefit period, after which the amount purchased with only USD 9 was consumed.

Results also suggested that the increased CVB leads to healthier options available for the household in general, suggesting that an increase in WIC CVB may have spillover impacts beyond the children on WIC. Pregnant and postpartum mothers eligible for WIC, although not the focus of the study, also reported benefits from an increased CVB amount. These women’s perceptions and experiences about their own CVB amount increase were in line with what was reported about the increase received for their children.

In addition, the findings showed that families were able to offset costs in other areas when the CVB amount was increased. For example, many families noted that they were able to buy more grains and meats for their family since they no longer had to pay out of pocket for more FV. For some households, the CVB increase also helped to alleviate the burden of increasing the cost of housing, transportation and overall food costs. Some families interviewed in this study emphasized they did not redeem the full value of the enhanced CVB because they could not get FV purchases to add precisely to the CVB amount, suggesting they were primarily focused on displacing their personal FV expenditures with the augmented CVB instead of increasing total FV purchases. Future analyses will be needed to explore whether the enhanced CVB was associated with increased FV intakes. Additionally, with the acceleration of inflation, families noted the rising cost of FV but stated the CVB increase allowed them the choice to spend on better quality and more variety while not having to worry so much about their budget.

When the monthly CVB amount was readjusted to USD 24, caregivers remained appreciative of the CVB amount greater than USD 9, but they noted that the previous amount of USD 35 was more consistent with their child’s dietary needs. The ideal mean CVB amount reported by the interviewed group was USD 38.17 [3,5]. While WIC is a food assistance program, the WIC food package was designed to be supplementary, meaning that the foods provided in the food package are only meant to partially meet a child’s dietary needs [19].

This study provides new information on WIC participants’ experiences and perceptions when food package changes occur. Interviews were conducted and analyzed by collaborating researchers not affiliated with WIC to minimize social desirability bias. Although interviews were conducted with a diverse sample based on race and ethnicity, one limitation of the study is that it included only participants from Los Angeles County in California where the cost of living is generally much higher than in other places. California ranks as the 4th most expensive state or federal district in the United States for the composite cost of living in 2021 [20]. Another limitation is that the sample was comprised of respondents to two previous surveys, thus they may be more satisfied with their WIC experience and open to sharing their experiences than non-respondents.

## 5. Conclusions

This study examined the perceptions and experiences of WIC families on the CVB amount change during the COVID-19 pandemic. Results suggest maintaining the CVB for children 1–4 years of age at the current level (USD 24/month), which is in line with the NASEM food package revision recommendations or increasing the amount further, will benefit WIC families’ ability to purchase and consume greater amounts, varieties and quality of FV. Study results provide timely perspectives to inform policymakers about benefit changes for the WIC food package. Based on the study participants’ positive perceptions of the CVB changes to date, a permanent increase in the CVB would be well-received by WIC participants and would increase access to fruits and vegetables among low-income households with young children in the United States. The permanent augmentation of the WIC CVB amount is also likely to increase WIC program participation and retention of eligible families, thus reducing the burden of access and leading to healthier diets supported by WIC.

## Figures and Tables

**Table 1 ijerph-19-10604-t001:** Characteristics of caregivers interviewed (n = 30).

Characteristic	Number of Participants or Mean (SD)
**Caregiver Race/Ethnicity**	
Hispanic	20
Non-Hispanic Black	10
**Household food insecure (in past month)**	17
**Number of children in household**	
1	8
2	12
3+	10
**Age of youngest child on WIC**	2.7 (0.8)
**Duration on WIC (years)**	5.8 (3.9)
**Frequency of grocery shopping (times/week)**	1.6 (1.3)
**Where usually shop for WIC foods ^1^**	
Large store (with multiple check out registers)	13
WIC only store	10
Both large and WIC stores	7
**Ease of finding fruit/vegetables wanted using WIC benefits ^2^**	
Very easy	26
Somewhat easy	4

^1^ Other answer options were small store with 1–2 registers and other. ^2^ Other answer options were somewhat hard and very hard.

**Table 2 ijerph-19-10604-t002:** Emergent themes for (A) experiences with CVB at USD 9 per month, (B) experiences after CVB increase to USD 35 per month, (C) other changes on spending outside of WIC benefits with CVB increase to USD 35 per month, (D) impacts of USD 35 per month CVB increase on dietary intakes and (E) experiences after CVB change to USD 24 per month ^1^.

Theme	Language	Representative Quotes
**1. Experiences with CVB at USD 9 per month**		
**1.1. Shopping for FV using WIC benefits is easy**	English	The (WIC) app tells you what FV are eligible, so it makes it a lot easier. They also have the WIC label next to some of the FV, like the kids fruit snacks. (English speaking participant with 4 years of WIC participation—EN17)
**1.2. CVB at USD 9 per month is inadequate**	English	Things are very expensive these days. So (USD) 9 does not go very far for produce, unfortunately. I try to catch all the sales, but it’s just with the inflation and the cost of living, or the cost of the produce itself, it (USD 9/month) does not go very far. I can buy frozen vegetables. Maybe get a nice amount of those that may carry me for most of the month. However, for fresh fruits, it was not enough. (English speaking participant with 3 years of WIC participation—EN10)
**2. Experiences after CVB increase to USD 35 per month**		
**2.1. Satisfaction increased with CVB increase to USD 35 per month**	English	I thought (the CVB increase) was amazing, because we eat more FV since there’s been an increase. (Before the increase) we were eating more processed food and I do not think that my daughter was getting adequate nutrition from eating those types of foods and she’s a picky eater too. So now that I am cooking vegetables with every dinner and we’re snacking on FV throughout the day. (English speaking participant with 2.5 years of WIC participation—EN4)
		It (CVB increase) gave us a better opportunity to eat more health conscious and make sure that the baby was getting all of the FV that he needs for each day. (English speaking participant with 2 years of WIC participation—EN11)
**Subtheme 1: Cost**	English	Sometimes (USD) 35 for vegetables is not enough because everything is pricey now. Because now you can get a little bit compared to before you get a lot. (English speaking participant with 8 years of WIC participation—EN7)
**Subtheme 2: Spending the full amount**	Both	By the fourth week of the month, I still had a balance, meaning that I had already gotten what I needed. So I felt like if I was to use it, the fruit would just go to waste. So I would be using something just because, instead of using it because I needed it… However, I also only have one child on WIC, so my family is not that big either. (English speaking participant with 7 years of WIC participation—EN6)
		It (CVB increase) was easier to choose from the variety that the supermarket had without thinking about what I am going to spend for a small amount. (Spanish speaking participant with 13 years of WIC participation—SP31)
**2.2. CVB increase resulted in purchasing greater amounts and variety of FV**	Both	I just did not feel pressured to go shopping and if I did not spend everything, the next time I went I could buy it and the fruit and everything would not spoil. (Spanish speaking participant with 4 years of WIC participation—SP28)
		I was not overwhelmed with having so much money just for produce where I felt like I needed to go spend that full amount in the month…. However, being that it was (USD) 35, it made it to where I was able to incorporate more of the fresh vegetables into my meals and actually plan a menu around it. Another benefit of it was that you were not predisposed to spend more on FV, which tends to go rotten faster, so I was actually able to use what I bought. (English speaking participant with 2.5 years of WIC participation—EN13)
		It (CVB increase) gave me more options to buy more and less worry to have... Okay if I needed to get it out of pocket am I going to have enough. So I do not have to worry about it for the month, so if I did not want to spend the entire (USD) 35 in one week or you know in one day it’s like okay... I am okay, I do not have to worry about having to come back next week because I am going to have extra left. Unlike with the (USD) 9,…I know it’s all going to go today. (English speaking participant with 8 years of WIC participation—EN19)
		Because before I was just looking for the most common fruits or most common vegetables and right now with the (increase) I could have access to other types of fruits …that she liked. (Spanish speaking participant with 4 years of WIC participation—SP28)
		It (CVB increase) was super good because we managed to buy more variety of FV and are able to make a salad or make other FV for the children. (Spanish speaking participant with 13 years of WIC participation—SP31)
		I started looking up different recipes for different vegetables because I would just buy the basics like carrots, celery, the same old stuff and make the same old soups (before the increase). So, I wanted to have more of a variety of what I was feeding my family. Furthermore, now I buy more cauliflower, squash. We eat a lot of roasted vegetables now and I just want to expose them to a bigger variety. (English speaking participant with 2.5 years of WIC participation—EN4)
		Oh it (CVB increase) definitely changed the amount and definitely changed the variety. I was able to buy her (child) different kinds of fruits, that she could then try. It opened up a bigger variety because I had more money to give her something other to try instead of just buying what I knew she liked, because that is all we had. (English speaking participant with 7 years of WIC participation—EN6)
		We were able to.... Look at… the whole scale of the produce section instead of just the sale section. (English speaking participant with 3 years of WIC participation—EN10)
		Because of the amount we’re getting (now), it’s easier to grab more without worrying... It’s three times the amount or more. So I do not worry about putting that money in our budget. (English speaking participant with 7 years of WIC participation—EN22)
**Subtheme 1: Variety of FV**	Both	Yes, I could buy more vegetables I had not tried. I tried other things, now she can know what they taste like and there’re more things that I can serve. (Spanish speaking participant with 4 years of WIC participation—SP28)
		I have struggled with her eating vegetables... Now (since the CVB increase) she likes to eat broccoli raw with like Ranch whereas before she would not try it at all and I think that has to do with her seeing me eat more vegetables that she wants to try it. So she’s definitely been trying new vegetables since I’ve been cooking more vegetables. (English speaking participant with 2.5 years of WIC participation—EN4)
		I am able to buy her a little more of what she likes and a little more of what she does not know about and introduce her to new FV. (English speaking participant with 7 years of WIC participation—EN6)
		It (USD 35/month) has afforded me to get a variety of things that I was not able to get before, and actually introduced my children to a lot more vegetables that were out of our reach (with USD 9/month). (English speaking participant with 3 years of WIC participation—EN10)
		I thought the change was extremely helpful. Because instead of just grabbing a few breakfast fruits, I was able…to try different fresh vegetables that normally I would not try. It (CVB increase) made me go more outside my comfort zone. It actually paid off because my son ate more of the newer things than the same old stuff he was used to. (English speaking participant with 2.5 years of WIC participation—EN13)
**Subtheme 2: Quality of FV**	Both	Before (the CVB increase), I grabbed the lowest quality and then with the (USD) 35 I can get better quality. (Spanish speaking participant with 3 years of WIC participation—SP26)
		I would say it (CVB increase) had a high impact. Because I have enough to buy the FV I need. Especially in the summer, you want to eat like a lot of fresh stuff. So I was really happy about the increase because I was able to make recipes that were good to eat during the summer. (English speaking participant with 2.5 years of WIC participation—EN4)
		Because it (CVB increase) was like a kid in the candy store. It’s like, wow, we have this resource now that we can get more and more. They (the children) just love... Especially the summertime fruits.... They were introduced to artichokes, and they wanted dragon fruit and things like that we were not able to do before. So it was it was great. (English speaking participant with 3 years of WIC participation—EN10)
		When it (CVB increase) happened during the summer months, there’s a lot of berries in season. So it was good to get the FV that were in season and when it was (USD) 9 I would use it to get Cuties and bananas, mostly, because that was what we would be able to get with (USD) 9. With (USD) 35, I could get strawberries, grapes, raspberries, blueberries, just more variety of fruits that my son likes to eat. (English speaking participant with 9 years of WIC participation—EN15)
**Subtheme 3: Offset increased cost of FV**	Both	Since blackberries are a bit more expensive, so with what WIC give us one takes advantage of grabbing what is usually more expensive in stores. (Spanish speaking participant with 5 years of WIC participation—SP30)
		Normally I do not buy a lot of FV because the prices are high… but I was able to do that with the WIC this time. (English speaking participant with 2.5 years of WIC participation—EN4)
		We did not worry about the sale prices at that point… having a little bit more to work with made it where you’re not budgeting as hard for the things that you like. (English speaking participant with 2 years of WIC participation—EN11)
		So like Brussel sprouts, or some of the vegetables that cost a little bit more. I was able to purchase those things with extra stretch of the dollar. (English speaking participant with 2.5 years of WIC participation—EN13)
**3. Other changes on spending outside of WIC benefits with CVB increase to USD 35 per month**		
**3.1. CVB increase impacts purchases of other groceries**	Both	I had the option to spend from my pocket what I had available for vegetables to buy meat, buy chicken, buy fish to change my son’s diet. (Spanish speaking participant with 7 years of WIC participation—SP27)
		A lot of the time, we buy cheaper proteins and stuff to have more money for FV. So I think that (a CVB increase) would help people buy better quality food. (English speaking participant with 2.5 years of WIC participation—EN4)
		Because we had the extra money for the FV, we were able to give maybe chicken or meats, pasta, something to compensate or you know to have a little more at the house. (English speaking participant with 12 years of WIC participation—EN8)
		Aside from FV, we were able to buy more meat and grains, like cereal and oatmeal and stuff like that. (English speaking participant with 2 years of WIC participation—EN11)
		It (the CVB increase) definitely had a role in the variety of foods that I could eat because I have more resources to afford them. (English speaking participant with 9 years of WIC participation—EN15)
		With that extra money…we can buy an extra pack of chicken or get fish that we cannot usually splurge on because we have to work that into the budget for that fruit and vegetable lump sum. (English speaking participant with 4 years of WIC participation—EN20)
**Subtheme 1: Reduced reliance on less healthy foods**	English	I did not used to juice, and have enough fruits left over. So now we do not buy a lot of sugar-filled drinks because we juice. (English speaking participant with 7 years of WIC participation—EN6)
		You tend to eat less processed food when you have more natural foods. (English speaking participant with 12 years of WIC participation—EN8)
		Normally I will give him pastries for breakfast, like a toaster strudel type of thing. However, once I had more FV, that is all he wanted. He’ll have a banana at the house, and I will even pack one for his lunch as well. So he was consuming way more (FV). (English speaking participant with 2.5 years of WIC participation—EN13)
		If I had it in cash (instead of the increased CVB), it would be different. Maybe I would spend it on something else for them. Sometimes they do also like juice or cookies and chips. (English speaking participant with 9 years of WIC participation—EN15)
**Subtheme 2: Reduced the burden of increased cost**	Both	More than anything (now), Furthermore expensive is fruit and vegetables. (Spanish speaking participant with 7 years of WIC participation—SP27)
		I think I’ve gotten a little bit more just in general, because of COVID, with the kids being home more....They would get a couple of more meals at school sometimes…., so I definitely increased the amount of food that we had in the house on hand for them. (English speaking participant with 9 years of WIC participation—EN15)
**C.2. CVB has minimal impact spending on other household needs**	Both	I’ve spent more on the car, gasoline, housing. They also raised my rent. (Spanish speaking participant with 19 years of WIC participation—SP32)
		I’ve been able to save a little bit more for transportation. As you know, gas prices have been a little bit higher than usual. So it’s been helping me maintain that. (English speaking participant with 3 years of WIC participation—EN5)
		(The cost of) everything got higher or stayed the same. The bills [got higher]. It’s cold right now, so the gas is getting high, and our electricity bill is kind of high. (English speaking participant with 4 years of WIC participation—EN17)
**4. Other changes on spending outside of WIC benefits with CVB increase to USD 35 per month**		
**4.1. Impact on diet of child**	Both	Both the amount and variety (increased) because with the price changes the variety. Because there are times I have to choose between prices and that is where the variety of FV changes. (Spanish speaking participant with 19 years of WIC participation—SP32)
		Because when you get more stuff, a lot more FV, you’re able to make different dishes and you can introduce other foods to her with the FV as well. (English speaking participant with 12 years of WIC participation—EN8)
		Yes, yes, yes (FV child ate increased), because I had more. Fortunately, I have really good eaters, so I they do not waste the food. Furthermore, because it’s new, they want to try it. They like it and so they consume more. I like to serve some FV with every meal and I cook at least three times a day. So we were going through a lot. Furthermore, I had that (FV) for their snacks as well. (English speaking participant with 3 years of WIC participation—EN10)
		Before I was sticking to apples, oranges and bananas, and then once it (CVB) went up, that is when I started getting more fresh broccoli, different salads and strawberries. (English speaking participant with 2.5 years of WIC participation—EN13)
		She (child) will pick the fruit over the other stuff that we have like over the cookie or chips or whatever. She sees the fruit first, the fruit is out and then the other stuff is more in the cabinet. So, she picks the fruit first because there’s more fruit (with the CVB increase). Instead of just in the beginning of the month, she got a few bananas and some grapes and then it’s gone. (English speaking participant with 10 years of WIC participation—EN14)
**4.2. CVB increase also has positive impacts on diet of others in household**	Both	Yes, it changed a little because you saved a little more money and that money that one saved was spent on other things. (Spanish speaking participant with 3 years of WIC participation—SP26)
		I started buying different varieties of vegetables, like for instance, now we buy cauliflower every week and I’ve been looking up different recipes, and it turns out the other people in my family really liked it as well, whereas before I just bought more simple and cheaper vegetables. (English speaking participant with 2.5 years of WIC participation—EN4)
		Well, it increased our (overall family’s) overall produce. We ate a lot more greens, a lot more of the FV, the summertime fruits… Especially during the summer, the kids were not able to go out too much, even though things were lifted. So a lot of times, of course everybody’s home, so everybody’s snacking more, because we are …nearer all the food. Therefore, to have the fresh fruits, the fresh vegetables, I can cut up... It helped, at least for me, to feel like they’re eating something healthy even though they’re eating all the time. (English speaking participant with 3 years of WIC participation—EN10)
		Me and my husband want to be health conscious, right? We want to do what is the healthiest thing, but when it comes to our situation, we put the baby first. So, when it was at (USD) 9 we were only making sure his snacks and his nutrition was taken care of. With the (USD) 35, we were able to branch out and then I was able to make sure my husband had his apricot and his peaches and stuff like that, things that he liked. So with the (USD) 35, it just expanded our horizons to be able to eat the things that we like as well. (English speaking participant with 2 years of WIC participation—EN11)
		Yes, I definitely ate more FV, too, because I do not like for them to go bad, so we will make sure we try to eat them up. (English speaking participant with 9 years of WIC participation—EN15)
**Subtheme 1: Cost of FV**	Spanish	I’ve bought a little more fruit. In the other things, it’s been the same because the price has also increased. We try not to exceed our expenses because if we do not do that, we will not have enough for rent. (Spanish speaking participant with 7 years of WIC participation—SP27)
**4.3. Other changes over the summer**	Both	The student EBT helped me stock up on fruits, vegetables, and meat. (Spanish speaking participant with 19 years of WIC participation—SP32)
		The increase in CalFresh, which was extremely helpful because everybody being home made it so that everyone was eating more, so the cost of groceries went up in the household. Furthermore, then being home and not working during the pandemic made it to where you did not have the extra dollar to spend on groceries. (English speaking participant with 2.5 years of WIC participation—EN13)
		With the EBT cards, the P-EBT cards it makes so much difference and it’s so helpful. I can buy as much as we want, as much as the kids want. They can make choices of what they want to eat. We are not limited to a certain amount in the grocery store. (English speaking participant with 5 years of WIC participation—EN16)
		The increase, the EBT increase, the WIC increase, made it better. It made the whole thing better. (English speaking participant with 3 years of WIC participation—EN18)
**5. Experiences after CVB change to USD 24 per month**		
**5.1. Monthly CVB amount of USD 24 is better than USD 9, but USD 35 is more in line with child’s needs**	Both	The (USD) 24 is better than (USD) 9 because with (USD) 9 it is not enough for anything. With (USD) 24, one can buy a week’s worth. (Spanish speaking participant with 19 years of WIC participation—SP32)
		(The USD 24) will be fine, because it’s better than the (USD) 9. I would not be that upset. That (USD) 9, I’ve never really understood why it was so little. What do they think? Do they think we do not eat fruits? You know, I was grateful for what it was. (English speaking participant with10 years of WIC participation—EN14)
		For me personally, I think it’s perfect. I am able to buy exactly what I want. I do not waste any money. I do not need any more. I am using exactly what they gave me, and I am able to buy exactly what I need for the whole month. (English speaking participant with 7 years of WIC participation—EN6)
		I think that it’s not enough, because we fall into the same thing that prices have been very high and we buy the same quantities that we bought before it rose to (USD) 35. I do not think they should have changed it, I think they should have just kept it at (USD) 35. (Spanish speaking participant with 7 years of WIC participation—SP27)
		Because in the end, we want our kids to eat more vegetables and fruits compared to eating the more affordable snack chips and stuff like that. (English speaking participant with 8 years of WIC participation—EN7)
**5.2. Positive impacts on grocery purchases**	Both	He (child) continues to eat FV. Not the same amount and not the same times like if I bought fruits more often like every week. Now I buy like every other week. The portions and quantities have changed, now he eats fewer portions, because since everything is more expensive, I buy less. (Spanish speaking participant with 19 years of WIC participation—SP32)
		It (USD 24/month) still allows me to broaden my scope a little bit more. Not as much as the (USD) 35 obviously, but a little bit more. I can pick and choose, and I can try something different. (English speaking participant with 3 years of WIC participation—EN10)
		It (the change to USD 24) was just a slight increase, and everything else went up (in cost). So, she’s kind of eating the same (as when the CVB was USD 9). Maybe an extra fruit? (English speaking participant with 3 years of WIC participation—EN21)
**Subtheme 1: Introduction of new FV to child’s diet**	English	Because it (USD 24/month) provides enough just for me to be getting FV that they (children on WIC) like, instead of getting things that they (children) do not like. (English speaking participant with 4 years of WIC participation—EN17)
**5.3. Optimal CVB amount for children is more than USD 9 or USD 24 per month**	English	Do not get me wrong, the (USD) 35 was amazing. What you give me is appreciated no matter what number it is. Does the increase in the WIC help my family? Definitely. However, I do not have a number that I feel that WIC should give me because I am just appreciative of whatever they give me if that makes sense. (English speaking participant with 2 years of WIC participation—EN11)
		I really like to (USD) 35, but anything more than (USD) 9 dollars is appreciated. It was better for meal prep and meal planning. I could get a lot out of it. Furthermore, it helps cover my vegetables that I needed for dinners and things like that. It provided snacks, which is what I prefer him (child) to eat over salty, sugary snacks. I rather him have fruit so if I can give him more healthy options then I would appreciate that. (English speaking participant with 9 years of WIC participation—EN15)

^1^ Alpha numeric codes at end of parentheticals in Table 2 indicate unique participant identifiers.

## Data Availability

The data include personal identifiable information and will not be publicly posted. Inquiries about the data can be sent to Catherine E. Martinez (catherinem@phfewic.org).

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
