# Peer review of "California WIC Participants Report Favorable Impacts of the COVID-Related Increase to the WIC Cash Value Benefit"

_ijerph, 2022, doi:10.3390/ijerph191710604_

Round 1
Reviewer 1 Report
This study examined the perceptions and experiences of WIC families on the CVB amount change during the COVID-19 pandemic. The topic is interesting. However, there are still some deficiencies, and this paper can be improved as follows:
1. Some terms should be clearly defined or explained, especially CVB, EN17, etc. More detailed information about how the policy of the WIC food package is implemented should be provided.
2. My major concern is that the sample is stratified by language (English and Spanish) and race (African American and non-African American), however, the authors did not discuss the possible differences between them. In the presentation of the results, the authors need more discussion on these differences. The writing of the abstract should be improved.
3. There are many abbreviations in the text, it is better to condense them into one table.
Reviewer 2 Report
Title: California WIC Participants Report Favorable Impacts of the COVID-related Increase to the WIC Cash Value Benefit
The manuscript presents a WIC participant perception of increased WIC Cash Value Benefits (CVB) for fresh produce during the COVID-19 pandemic. The paper is interesting from qualitative point of view although the focus of the study is limited. I suggest the following issues to be addressed to improve the manuscript.
· It’s not clear why authors selected/categorized African American (Black) and Non African American from English speaking category. The sampling procedure is biased towards AA ethnic group. There is no representation from non-AA group. Study would have been meaningful if the other key other groups (White, Asian, Pacific Islanders etc.) were included/considered in the study given the diversity in CA. A clear explanation on exclusion of important ethnic groups is needed.
· The number of daily servings of F&V is an important parameter for the well-being of people. This information is missing in the study. If authors have information related to changes in daily servings of F&V during their interviews, I strongly suggest including this information in the manuscript.
· Authors have mentioned that the study findings would be useful for policy decisions. It would be helpful if authors could present specific policy recommendations based on study findings.
Reviewer 3 Report
The manuscript by Martinez, C. E. et al. entitled “California WIC Participants Report Favorable Impacts of the COVID-related Increase to the WIC Cash Value Benefit” is a study with an aim to examine the satisfaction of the beneficiaries of the WIC program with CVB changes caused by the pandemic and inflation.
In the manuscript one part of the methodology is not clearly described so some parts were more difficult to understand.
English proofread by native speaker is needed for clarity of the manuscript.
In general, the discussion part is flawed by the lack of comparisons with similar research and the references are deficient and mainly refer to the legislative framework.
Some technical comments:
In the abstract it is not it necessary to provide subtitles so you can remove it.
Row 17 – with caregivers with children? – please rewrite the sentence
Conclusion in the abstract – inform policy – did you mean policy makers?
For health benefits there are
Purpose of the study – please shorten that part or write it more clearly
Row 149-150 - which was the way for choosing the presented answers? Also, in the part 3.4.1. and 3.4.2. same and complete answers from table 2 were presented
Under the “theme category“ in the Result part, as well as through the manuscript, it is not clearly defined that the increase to $35 was followed by the increase to $24 and that the satisfaction with both changes was examined
In discussion (row 256) “two different changes” - does this represent two different financial aid amounts ($35 and $24)
The conclusion part is written in a very general and the amount of aid is highlighted, which is almost not represented in the manuscript.
Row 321? – In row 149/150 you state that Table 2 is part of the supplementary material.
Row 351 – reference No 5 – Dietary Guidelines valid are those for period 2020-2025 released in 2020 – that is yours reference No 12
Row 360/361 - Leibtag, Ephraim, and Aylin Kumcu. The WIC Fruit and Vegetable Cash Voucher: Does Regional Price Variation Affect Buying Power? EIB-75. U.S. Dept. of Agriculture, Econ. Res. Serv. May 2011.
Round 2
Reviewer 3 Report
I would like to thank the authors for their efforts to improve the manuscript.
Some rearrangements have been made to the text. The paper is now somewhat clearer.
But there are still some things that should be clarified or changed.
For example, in Abstract – line 24 - Findings are expected to…. changes to the WIC food package…. (what with the changes?? – make changes, or?)
Primary purpose of this study was well supported and the results mainly come down to that part. Results that should explain the second purpose are not clearly point out.
The conclusion should also be constructed based on the primary and secondary objective of this study
e.g. “A policy implication of these results is that permanent augmentation of the CVB, in line with the National Academies of Science, Engineering, and Medicine food package revision recommendations, would be both well-received by participants and increase access to fruits and vegetables among low-income households with young children in the United States” – true but still there is no clear, concrete conclusion
